# Evolution Strategies at Scale: LLM Fine-Tuning Beyond Reinforcement Learning

## Abstract

Fine-tuning pre-trained large language models (LLMs) for down-stream tasks is a critical step in the AI deployment pipeline. Reinforcement learning (RL) is arguably the most prominent fine-tuning method, contributing to the birth of many state-of-the-art LLMs. In contrast, evolution strategies (ES), which once showed comparable performance to RL on models with a few million parameters, was neglected due to the pessimistic perception of its scalability to larger models. In this work, we report the first successful attempt to scale up ES for fine-tuning the full parameters of LLMs, showing the surprising fact that ES can search efficiently over billions of parameters and outperform existing RL fine-tuning methods in multiple respects, including sample efficiency, tolerance to long-horizon rewards, robustness to different base LLMs, less tendency to reward hacking, and more stable performance across runs. It therefore serves as a basis to unlock a new direction in LLM fine-tuning beyond what current RL techniques provide.

## 1 Introduction

The rapid development of more capable large language models (LLMs; Touvron et al., 2023; Achiam et al., 2024; AI@Meta, 2024; Jiang et al., 2024; Liu et al., 2024; Anthropic, 2025; Google, 2025) has made many scientific and engineering domains amenable to AI-based automation (Singhal et al., 2023; Wu et al., 2023; Rozière et al., 2024; Romera-Paredes et al., 2024). As a result, fine-tuning the pre-trained models to accommodate specific tasks and to improve alignment with user preferences has become an important part of the LLM deployment pipeline (Ouyang et al., 2022; Rafailov et al., 2023; Latif & Zhai, 2024; Guo et al., 2025a). Reinforcement learning (RL) is currently the pre-dominant choice for such fine-tuning (Ouyang et al., 2022; Bai et al., 2022; Shao et al., 2024; Guo et al., 2025a;b; Srivastava & Aggarwal, 2025). Several challenges have emerged: First, RL methods incur low sample efficiency (Vemula et al., 2019) and high variance of the gradient estimator (Salimans et al., 2017; Sutton & Barto, 2018) when handling long-horizon rewards, which is a common case for LLM fine-tuning with outcome-only rewards. Proper credit assignment at token level for RL fine-tuning methods is difficult (Zhang et al., 2025; Song et al., 2025; Guo et al., 2025b) and possibly unhelpful (Uesato et al., 2022; Jia et al., 2025; Guo et al., 2025b). Second, RL techniques are sensitive to the choice of base LLMs, resulting in inconsistent fine-tuning performance across different models (Gandhi et al., 2025). Third, RL techniques have an inherent tendency to hack the reward function, leading to undesirable behaviors (Gao et al., 2023; Denison et al., 2024; Fu et al., 2025). Fourth, RL fine-tuning is often unstable across multiple runs even with the same parametric setup, increasing the cost of fine-tuning significantly (Choshen et al., 2020; Zhong et al., 2025).

Evolution Strategies (ES), a class of population-based zeroth-order optimization algorithms, is a possible alternative. ES has several unique advantages over RL in traditional control and gaming problems: it is highly parallel, tolerates long-horizon rewards well, explores extensively, needs less computation (no backpropagation), and is robust to setup parameters (Salimans et al., 2017; Chrabaszcz et al., 2018; Conti et al., 2018). However, ES has received much less attention than RL during the LLM era. Standard ES works by searching and optimizing in the original parameter space directly, for which the dimension was no more than a few million in past implementations (Salimans et al., 2017; Zhang et al., 2017; Lehman et al., 2018; Lorenc & Neruda, 2025). It was assumed that if the model is very large, it is significantly more difficult and sample-inefficient to explore in parameter space compared to action-space (Vemula et al., 2019). The number of parameters in LLMs is usually on the order of billions, which may seem infeasible for ES to directly tackle. Exist-

ing workarounds include applying ES only to the last layer of the original model (Toledano-López et al., 2022), using ES to fine-tune a lower-dimensional adapter (Jin et al., 2024), and searching in action space as in standard RL (Huang et al., 2025). Directly searching in the full parameter space of LLMs has remained a daunting challenge.

This paper is aimed at meeting this challenge. For the first time, ES is scaled to multi-billion-parameter search spaces, by searching directly over the full parameter space of LLMs in fine-tuning tasks. The approach is based on a memory-efficient implementation of an algorithmically simplified ES variant, with support for parallelization within and across GPUs. Performance is compared with state-of-the-art (SOTA) RL methods in fine-tuning various LLMs in a standard reasoning benchmark task, and behavioral differences from RL are analyzed in terms of fine-tuning for conciseness. This version of ES was able to search directly over billions of parameters, and exhibit surprisingly good fine-tuning performance compared to RL methods in multiple aspects:

- ES only needs response-level rewards, making it a perfect fit for fine-tuning on reasoning tasks that have only sparse long-horizon outcome rewards. In particular, ES obtained significantly better fine-tuned models than RL in the Countdown task with such rewards

- Counterintuitively, even though ES explores in the parameter space with billions of parameters, it is more sample efficient than RL methods that explore in the action space, which is much smaller. Further, ES was able to find good solutions with a population size of only 30. As a comparison, previous ES implementations (Salimans et al., 2017; Zhang et al., 2017; Lehman et al., 2018; Lorenc & Neruda, 2025) utilized a population size of 10,000 or more with much smaller models (i.e. millions of parameters or less).

- ES is significantly more robust than RL across different LLMs. While RL fine-tuning failed on some LLMs, ES provided good fine-tuning for all of them. ES benefits from its exploration in parameter space, making it less sensitive to initial states of the LLMs.

- Whereas RL tends to hack the reward function if no other penalty is added, ES consistently maintains reasonable behaviors during fine-tuning. The main reason is that ES optimizes a solution distribution (Lehman et al., 2018), which is more difficult to hack, while RL optimizes a single solution.

- ES's behavior is more consistent than RL's across different runs. This property can significantly reduce expected cost of fine-tuning.

- Fine-tuning with ES is based on inference, and therefore no backpropagation calculations are needed. A significant amount of GPU memory can therefore be saved (Malladi et al., 2023).

The study reported in this paper serves as a first step in demonstrating the potential of ES for fine-tuning LLMs. The surprising and counterintuitive findings motivates further work scaling up ES to larger LLM fine-tuning tasks. Given the unique advantages over the state of the art, ES opens up new opportunities in parameter-space exploration, outcome-only fine-tuning, and large-scale distributed post-training.

## 2 RELATED WORK

Evolution Strategies (ES, Rechenberg, 1973; Schwefel, 1977) are a class of evolutionary algorithms (EAs) for solving numerical optimization problems. The main idea is to sample a population of solutions through perturbations, then recombine the perturbed solutions based on their fitness values to form the population for the next generation. This process repeats until a termination condition is triggered, e.g., the maximum number of generations is reached. Among the different variants of ES, CMA-ES (Hansen & Ostermeier, 2001), which utilizes a multivariate Gaussian distribution with full covariance matrix to sample the population, and natural ES (Wierstra et al., 2008; 2014), which uses natural gradient to guide the search, are two popular methods for traditional optimization problems. Although ES has long been used to evolve parameters of neural networks (NNs), (Igel, 2003), Salimans et al. (2017) were the first to scale the approach up to deep learning networks. Comparable performance to RL methods in control and gaming environments was observed, and several unique advantages of ES highlighted. This seminal work paved the way for several follow-up studies. Zhang et al. (2017) used ES to optimize a convolutional NN with around three million

parameters. They found that with a large enough population size, ES can approximate the performance of traditional stochastic gradient descent (SGD). Lehman et al. (2018) further optimized an NN comprising nearly 167,000 parameters with both ES and a traditional finite-difference (FD) gradient estimator. Because ES optimizes the average reward for the entire population, whereas FD optimizes the reward for a single solution, it obtained models that were more robust to parameter perturbations. Lorenc & Neruda (2025) applied ES to optimize decision transformers in RL environments, and observed promising results for model sizes up to around 2.5 million parameters. In a related study, another traditional EA, namely genetic algorithm (GA) with mutations only, was extended to a high-dimensional space (Such et al., 2017). Encouraging results were observed in different types of models with up to around four million parameters (Such et al., 2017; Risi & Stanley, 2019). However, although these studies were promising, the scale of these implementations was still significantly less than the size of current LLMs.

Synergies between Evolutionary Algorithms (EAs) and LLMs have received increasing attention in recent years (Wang et al., 2025; Wu et al., 2025). Popular research directions include EAs for prompt optimization (Sun et al., 2022b;a; Zhao et al., 2023; Guo et al., 2024), utilizing LLMs as evolutionary operators (Meyerson et al., 2024; Lehman et al., 2024; Romera-Paredes et al., 2024; Novikov et al., 2025), and merging LLMs through evolution (Du et al., 2024; Akiba et al., 2025). Applying EAs to optimize billions of parameters in LLMs is generally perceived to be intractable, but a few studies have been successful at a smaller scale. For example, Toledano-López et al. (2022) fine-tuned the last layer (with 325 parameters) of an mT5-based transformer via CMA-ES. Jin et al. (2024) optimized the low-rank adapter parameters (with dimensionality up to 1600) using CMA-ES and the Fireworks algorithm. Sanchez Carmona et al. (2024) applied a GA to fine-tune around 9.5 million parameters of a transformer encoder, though poorer performance than the traditional Adam optimizer was observed. Huang et al. (2025) proposed a hybrid algorithm that performs exploration in action space instead of parameter space, and it was only used in the final epoch of supervised fine-tuning (SFT). The work in this paper significantly extends this prior research by successfully scaling ES to search in the billions of parameters of LLMs, leading to surprisingly good fine-tuning performance.

Fine-tuning using RL is a critical step during the training of many landmark LLMs (Ouyang et al., 2022; Bai et al., 2022; Shao et al., 2024; Guo et al., 2025a;b). Proximal Policy Optimization (PPO; Schulman et al., 2017) and Group Relative Policy Optimization (GRPO; Shao et al., 2024) are the two predominant methods. PPO introduces a clipped surrogate objective to limit the update scale in each step with respect to the old policy, and it usually works with a value model in an actor-critic manner. GRPO simplifies the pipeline of PPO by replacing the value model with group advantage, which is calculated based on direct evaluations of multiple responses. As discussed in Section 1, in the context of LLM fine-tuning, these methods struggle with several fundamental limitations, including the dilemma in handling long-horizon reward (Vemula et al., 2019; Salimans et al., 2017; Zhang et al., 2025; Song et al., 2025; Uesato et al., 2022; Jia et al., 2025; Guo et al., 2025b), sensitivity to base LLMs (Gandhi et al., 2025), tendency to hack reward (Gao et al., 2023; Denison et al., 2024; Fu et al., 2025), and instability across runs (Choshen et al., 2020; Zhong et al., 2025). ES inherently avoids these limitations, leading to better fine-tuning performance.

Existing RL fine-tuning methods are overwhelmingly based on action-space exploration. Parameter space exploration has received much less attention, though some such studies do exist (Rückstieß et al., 2008; Sehnke et al., 2010; Rückstieß et al., 2010; Plappert et al., 2018). Although promising performance was observed in problems with sparse rewards, the scale of the tested models was far smaller than that of LLMs. Vemula et al. (2019) performed a theoretical analysis of different exploration strategies, and found that the complexity of the parameter space exploration increased quadratically with the number of parameters, whereas the complexity of action space exploration depended on action dimensionality quadratically and horizon length of the reward quartically. Based on the classical SPSA optimization method (Spall, 1992), Malladi et al. (2023) proposed a zeroth-order optimizer MeZO that directly worked in parameter space for fine-tuning LLMs. MeZO significantly reduced memory requirements, but its fine-tuning performance was no better than other baselines. In contrast, the ES implementation in this paper performs exploration in multi-billion-parameter search spaces, and outperforms all baselines.

---

**Algorithm 1** Basic ES Algorithm

---

**Require:** Pretrained LLM with initial parameters $\boldsymbol{\theta}_0$, reward function $R(\cdot)$, total iterations $T$, population size $N$, noise scale $\sigma$, learning rate $\alpha$.

1: **for** $t = 1$ to $T$ **do**                                                 $\triangleright$ outer ES iterations
2:     **for** $n = 1$ to $N$ **do**
3:         Sample noise $\boldsymbol{\varepsilon}_n \sim \mathcal{N}(0, \boldsymbol{I})$
4:         Compute reward for perturbed parameters $R_n = R(\boldsymbol{\theta}_{t-1} + \sigma \cdot \boldsymbol{\varepsilon}_n)$
5:     **end for**
6:     Normalize $R_n$
7:     Update model parameters as $\boldsymbol{\theta}_t \leftarrow \boldsymbol{\theta}_{t-1} + \alpha \cdot \frac{1}{N} \sum_{n=1}^{N} R_n \boldsymbol{\varepsilon}_n$
8: **end for**

---

## 3 METHOD

This section introduces the basic algorithmic structure of ES, followed by a detailed description of its implementation for LLM fine-tuning.

### 3.1 BASIC ES ALGORITHM

The ES implementation in this paper is an algorithmically simplified variant of natural evolution strategies (NES) (Wierstra et al., 2008; 2014). The overall design is similar to OpenAI ES (Salimans et al., 2017), which simplified NES with fixed covariance for perturbation noise.

Given a pretrained LLM with initial parameters $\boldsymbol{\theta}_0$ and a target reward function $R(\cdot)$, the task is to fine-tune the parameters so that the reward function is optimized (Algorithm 1). In each iteration, $N$ perturbed models are sampled by adding random Gaussian noise $\boldsymbol{\varepsilon}_n$ to their parameters. The noise is i.i.d. in each dimension of the parameter space, and it is scaled by the hyperparameter $\sigma$. The perturbed models are evaluated to obtain their reward scores $R_n$. The final update of the model parameters aggregates the sampled perturbations by weighting them using their normalized reward scores. The standard update equation $\boldsymbol{\theta}_t \leftarrow \boldsymbol{\theta}_{t-1} + \alpha \cdot \frac{1}{\sigma} \frac{1}{N} \sum_{n=1}^{N} R_n \boldsymbol{\varepsilon}_n$ is simplified to $\boldsymbol{\theta}_t \leftarrow \boldsymbol{\theta}_{t-1} + \alpha \cdot \frac{1}{N} \sum_{n=1}^{N} R_n \boldsymbol{\varepsilon}_n$ by digesting the term $\frac{1}{\sigma}$ into the learning rate $\alpha$.

To improve scalability, a number of modifications to this basic algorithm were made as detailed in the next section.

### 3.2 IMPLEMENTATION DETAILS

Algorithm 2, the actual implementation of ES for this paper, expands on the above algorithm in seven ways:

(1) **Noise retrieval with random seeds:** Similar to Salimans et al. (2017); Such et al. (2017), only the random seeds are stored to reduce GPU memory usage. The perturbation noise used during sampling can be retrieved exactly by resetting the random number generator with specific random seeds. (2) **Parallel evaluations:** In each iteration, the perturbed models can be evaluated fully in parallel by assigning a separate random seed to each process. (3) **Layer-level in-place perturbation and restoration:** To reduce the peak GPU memory usage, the model parameters are perturbed in-place layer by layer, with corresponding random seeds archived. After evaluation of the perturbed model, the model parameters are restored by subtracting the same noise perturbations using the archived random seeds. For each evaluation process, apart from the model parameters, the only additional memory needed is to store a tensor the size of a layer temporarily. (4) **Reward normalization:** The rewards of the perturbed models are normalized using $z$-score within each iteration, so that the normalized rewards for each iteration have a mean of 0 and standard deviation of 1. This normalization makes the reward scale consistent across iterations and tasks. (5) **Greedy decoding:** The perturbed models use greedy decoding to generate the responses for reward evaluations. As a result, the perturbed models are evaluated deterministically, so that all performance differences come from the exploration in parameter space instead of action space. (6) **Decomposition of the parameter update:** At the end of each iteration, the aggregated update of model parameters is performed in-place in a decomposed manner, gradually adding up layer by layer and seed by seed, significantly

---

**Algorithm 2** ES Implementation for LLM Fine-Tuning

---

**Require:** Pretrained LLM with initial parameters $\boldsymbol{\theta}_0$, reward function $R(\cdot)$, total iterations $T$, population size $N$, noise scale $\sigma$, learning rate $\alpha$, number of parallel process $P$.

1: Create $P$ processes, each instantiates a model with the same initial parameters $\boldsymbol{\theta}_0$, with one process as the main process

2: **for** $t = 1$ to $T$ **do** ▷ ES iterations

3:     Sample N random seeds $s_1, s_2, \ldots, s_N$

4:     Assign random seeds to $P$ processes

5:     **for** $n = 1$ to $N$ **do**

6:         For the process handling $s_n$, reset its random number generator using random seed $s_n$

7:         **for each LLM layer do** ▷ perturbation within current process

8:             Sample noise $\boldsymbol{\varepsilon}_{n,l} \sim \mathcal{N}(0, \boldsymbol{I})$, which has the same shape as the $l$th layer's parameters

9:             Perturb the $l$th layer's parameters in-place: $\boldsymbol{\theta}_{t-1,l} \leftarrow \boldsymbol{\theta}_{t-1,l} + \sigma \cdot \boldsymbol{\varepsilon}_{n,l}$

10:         **end for**

11:         Compute reward for perturbed parameters $R_n = R(\boldsymbol{\theta}_{t-1})$ ▷ within current process

12:         For the process handling $s_n$, reset its random number generator using random seed $s_n$

13:         **for each LLM layer do** ▷ restoration within current process

14:             Sample noise $\boldsymbol{\varepsilon}_{n,l} \sim \mathcal{N}(0, \boldsymbol{I})$, which has the same shape as the $l$th layer's parameters

15:             Restore the $l$th layer's parameters in-place: $\boldsymbol{\theta}_{t-1,l} \leftarrow \boldsymbol{\theta}_{t-1,l} - \sigma \cdot \boldsymbol{\varepsilon}_{n,l}$

16:         **end for**

17:     **end for**

18:     Normalize the reward scores by calculating the $z$-score for each $R_n$: $Z_n = \frac{R_n - R_{\mathrm{mean}}}{R_{\mathrm{std}}}$, where $R_{\mathrm{mean}}$ and $R_{\mathrm{std}}$ are the mean and standard deviation of $R_1, R_2, \ldots, R_N$.

19:     **for** $n = 1$ to $N$ **do** ▷ in main process only

20:         Reset current random number generator using random seed $s_n$

21:         **for each LLM layer do**

22:             Sample noise $\boldsymbol{\varepsilon}_{n,l} \sim \mathcal{N}(0, \boldsymbol{I})$, which has the same shape as the $l$th layer's parameters

23:             Update $l$th layer's parameters in-place as $\boldsymbol{\theta}_{t,l} \leftarrow \boldsymbol{\theta}_{t-1,l} + \alpha \cdot \frac{1}{N} Z_n \boldsymbol{\varepsilon}_{n,l}$

24:         **end for**

25:     **end for**

26:     Update the model parameters of all processes to $\boldsymbol{\theta}_t$

27: **end for**

---

reducing the peak GPU memory needed. (7) **Learning rate digestion:** The standard update equation $\boldsymbol{\theta}_t \leftarrow \boldsymbol{\theta}_{t-1} + \alpha \cdot \frac{1}{\sigma} \frac{1}{N} \sum_{n=1}^{N} R_n \boldsymbol{\varepsilon}_n$ is simplified to $\boldsymbol{\theta}_t \leftarrow \boldsymbol{\theta}_{t-1} + \alpha \cdot \frac{1}{N} \sum_{n=1}^{N} R_n \boldsymbol{\varepsilon}_n$ by digesting the term $\frac{1}{\sigma}$ into the learning rate $\alpha$, simplifying the computation and parametric setup.

In order to keep the algorithm simple, common enhancements in OpenAI ES Salimans et al. (2017) such as rank transformation of rewards (Wierstra et al., 2014), mirrored sampling (Sehnke et al., 2010), weight decay, and virtual batch normalization (Salimans et al., 2016) are not used, and neither are more advanced optimizers like Adam (Kingma & Ba, 2015). They can be included in to improve results in future work.

## 4 EMPIRICAL STUDIES

This section first compares the fine-tuning performance of ES, PPO and GRPO on a standard reasoning benchmark. After that, behavioral differences between ES and RL are investigated in fine-tuning for conciseness in the next section.

### 4.1 PERFORMANCE IN THE COUNTDOWN TASK

Fine-tuning performance was measured in the Countdown task (Pan et al., 2025; Goodfellow et al., 2016), a symbolic reasoning benchmark, showing that ES is accurate and sample efficent across different kinds and sizes of LLMs, even when the RL approaches is not.

| Base Model | Original | RL | | | ES (ours) |
|---|---|---|---|---|---|
| | | PPO | GRPO (8) | GRPO (30) | |
| Qwen-2.5-0.5B-Instruct | 0.1 | 0.3 | 0.3 | 0.5 | **14.4** |
| Qwen-2.5-1.5B-Instruct | 0.7 | 14.2 | 13.9 | 14.8 | **37.3** |
| Qwen-2.5-3B-Instruct | 10.0 | 20.1 | 30.9 | 32.5 | **60.5** |
| Qwen-2.5-7B-Instruct | 31.2 | 55.1 | 54.2 | 52.8 | **66.8** |
| LLaMA-3.2-1B-Instruct | 0.4 | 11.2 | 14.5 | 13.0 | **16.8** |
| LLaMA-3.2-3B-Instruct | 3.2 | 35.3 | 39.4 | 38.8 | **51.6** |
| LLaMA-3.1-8B-Instruct | 8.1 | 42.8 | 49.9 | 51.3 | **61.2** |

Table 1: Accuracy (%) on the Countdown task across model families, sizes, and fine-tuning algorithms. Different model families are shaded for clarity; *Original* refers to directly evaluating the base model without any fine-tuning, and GRPO (8) and GRPO (30) indicate group sizes of 8 and 30. The same hyperparameters were used for all ES runs; a separate grid search for the best hyperparameters was run for each RL experiment.

**Countdown task.** The Countdown task (Pan et al., 2025; Goodfellow et al., 2016) requires constructing an arithmetic expression from a given set of numbers using basic operations $(+, -, \times, \div)$ to match a target value. For instance, the target 950 can be obtained from $\{100, 50, 6, 3\}$ with $100 \times (6 + 3) + 50 = 950)$. This constitutes a compact test of constrained symbolic reasoning, i.e. an important use case for fine-tuning.

**Experimental Setup.** A single fixed set of hyperparameters was used for all ES experiments. For RL, a separate hyperparameter sweep was done for each experiment. RL methods turned out sensitive to hyperparameters, in particular the KL-divergence penalty coefficient $\beta$ and learning rate $\alpha$, and did not make much progress if they were not set precisely. To mitigate this issue, for each model, a small grid of $\beta$ and $\alpha$ values were tested and the best-performing configuration selected (further details are provided in Table 3 in the Appendix A.1). This approach makes the comparison conservative with respect to ES, but it also highlights its robustness.

**ES improves upon PPO and GRPO across all tested models.** Previously, Gandhi et al. (2025) found that RL does not generalize well across models on the Countdown task. Table 1 confirms this result, and also demonstrates that ES does not have this problem. With each model in the Qwen2.5 family (0.5B–7B) and the LLaMA3 family (1B–8B), ES substantially improved over both PPO and GRPO, often by a large margin. Averaged across all models, ES improves over the base model by 36.4%, compared to 17.9% for PPO and 21.3% for GRPO with group size $N = 8$ and 21.4% for group size $N = 30$ (see Figure 5 in Appendix A.4 for a model-wise visual comparison). These results demonstrate that ES scales effectively across different model types and sizes, and does so significantly better than RL.

**ES is more sample efficient than RL.** Surprisingly, even when searching in a space with billions of parameters, ES is more sample efficient than the RL methods. The experiments reported in Table 1 were all run with the same total number of training sample evaluations (each training sample is one Countdown question). Figure 6 in Appendix A.4 further shows the learning curves; to reach the same level of fine-tuning performance as RL, ES needs less than 20% of the training sample evaluations in most cases. This observation is different from the that of Salimans et al. (2017), who found worse sample efficiency compared to RL. One critical factor is the population size $N$; the experiments in this paper had $N = 30$, whereas Salimans et al. (2017) used $N = 10,000$, which may explain the difference.

**ES is effective on smaller models.** Prior work on DeepSeek-R1 (Guo et al., 2025b) and Tiny-Zero (Pan et al., 2025) pointed out a key limitation of PPO and GRPO: they require sufficient large base models to improve. For instance, Pan et al. (2025) note that "for Qwen2.5-0.5B base, we know it fails to learn reasoning." Surprisingly, ES overcomes this limitation. As shown in Table 1, while PPO and GRPO indeed obtain only 0.3% accuracy on that model, ES boosts accuracy to 14.4%, thus eliciting reasoning even from the smallest-scale base model. This performance difference demon-

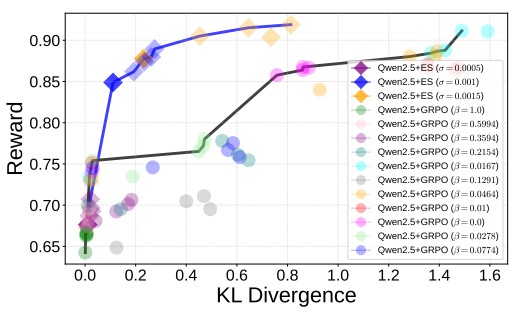

Figure 1: Mean conciseness reward and mean KL divergence from the base model for each fine-tuning checkpoint across different learning parameters. The Pareto front of ES (blue line) is higher and to the left of the GRPO Pareto front (black line) models, indicating that it found better tradeoffs. ES discovers these solutions without any KL divergence penalty, suggesting that it represents a distinctly different fine-tuning mechanism from the GRPO.

strates the benefit of parameter-space exploration in ES: while RL cannot find better actions from the limited initial model to bootstrap learning, ES modifies the model directly by adding perturbations in parameter space, possibly creating better models to facilitate further exploration. These results highlight a distinct advantage of ES: it is able to improve behavior even with smaller, weaker base models, thus expanding the scope of fine-tuning.

## 4.2 Behavioral Differences between ES and RL in fine-tuning for conciseness

In order to characterize the different approaches that ES and RL take, they were used to fine-tune Qwen-2.5-7B Instruct, towards more concise responses in question-answering. That is, fine-tuning was rewarded based on how concise the answers were, but not directly rewarded for its question-answering performance. In this setup, it was possible to analyze not only whether fine-tuning was effective, but also how it was achieved, including what its side effects were.

**Conciseness task.** For conciseness fine-tuning, a dataset of prompts $\mathcal{D} = \{x_1, .., x_K\}$, with a set of verifiable solutions $\{s_1, ..., s_K\}$, i.e. shortest possible correct answers, was used. For example, for the prompt "Name one primary color", possible shortest verifiable solution used is "Red". Following this approach, for each prompt $x \in \mathcal{D}$, the model was encouraged to generate a concise response $y$. To fine-tune the model to generate concise responses, a reward computed using the absolute length difference between the generated response $y$ and the corresponding verified solution $s_k$ was given to the model for each prompt $x_k$. The reward function $R$ for conciseness was defined as $R = -|\text{len}(y) - \text{len}(s_k)|$, where $\text{len}(\cdot)$ denotes the string length.

**Behavior metrics.** Behavior of the fine-tuned models was measured in two ways: the mean conciseness reward and the mean KL divergence from the base model (after Rafailov et al., 2023). KL divergence is useful as a proxy for the preservation of the base model's behavior. It correlates strongly with the question-answering performance of the model, but also conveys more information, i.e. the extent of the fine-tuning changes. A low KL divergence thus suggests that the fine-tuned model has not forgotten capabilities learned during pre-training. Further, as KL divergence increases, these capabilities are likely to break. Therefore, fine-tuning behavior can be characterized using the tradeoffs between reward and KL divergence. To compute the metrics, each fine-tuned model was evaluated on a set of held-out test prompts, with 20 responses sampled per prompt. The reward was computed using the model-generated response and the verifiable solution provided in the test dataset. The KL divergence between a fine-tuned model $\theta_{\text{FT}}$ and a base model $\theta_{\text{BASE}}$ for a given prompt $x$ and corresponding response $y$ was approximated following Schulman (2020) as

$$\text{KL}[\theta_{\text{FT}} \| \theta_{\text{BASE}}] = \frac{\theta_{\text{BASE}}(y_{i,t} \mid x, y_{i,<t})}{\theta_{\text{FT}}(y_{i,t} \mid x, y_{i,<t})} - \log \frac{\theta_{\text{BASE}}(y_{i,t} \mid x, y_{i,<t})}{\theta_{\text{FT}}(y_{i,t} \mid x, y_{i,<t})} - 1. \tag{1}$$

**ES discovers a dominant Pareto front.** Similarly to Rafailov et al. (2023), a Pareto frontier analysis was used to compare ES and GRPO, with mean reward and mean KL divergence as the metrics (Figure 1). The experimental setup is described in Appendix A.1. The ES Pareto front is represented by a blue line on top and the GRPO Pareto front by the black line below. That is, ES produced better tradeoffs than GRPO, i.e. models with higher reward and lower KL divergence. The GRPO results were achieved only after augmenting the conciseness reward with a KL divergence penalty (weighted by a parameter $\beta$). Without it, fine-tuning resulted in excessive divergence and incorrect answers. Remarkably, ES achieved superior tradeoffs without any KL divergence penalty,

| Model | $\beta$ | $\alpha$ | $\sigma$ | Reward $\uparrow$ | KL $\downarrow$ |
|---|---|---|---|---|---|
| Qwen-2.5-7B+GRPO | 0.0 | $5 \times 10^{-6}$ | ✗ | $0.867 \pm 0.054^*$ | $0.861 \pm 0.614^*$ |
| Qwen-2.5-7B+GRPO | 0.01 | $5 \times 10^{-6}$ | ✗ | $0.871 \pm 0.060^*$ | $1.354 \pm 0.873^*$ |
| Qwen-2.5-7B+GRPO | 0.0167 | $5 \times 10^{-6}$ | ✗ | $0.911 \pm 0.038$ | $1.591 \pm 0.811$ |
| Qwen-2.5-7B+GRPO | 0.0464 | $5 \times 10^{-6}$ | ✗ | $0.881 \pm 0.062$ | $1.384 \pm 1.187$ |
| Qwen-2.5-7B+ES | ✗ | 0.0005 | 0.001 | $0.889 \pm \mathbf{0.004}$ | $0.274 \pm \mathbf{0.096}$ |
| Qwen-2.5-7B+ES | ✗ | 0.00075 | 0.0015 | $0.919 \pm \mathbf{0.008}$ | $0.813 \pm \mathbf{0.212}$ |

Table 2: Behavior or GRPO and ES in terms of mean conciseness reward and mean KL divergence. The label $^*$ indicates cases where reward hacking was observed. Only models that did not hack the reward were included in the results.

suggesting that ES fine-tuning is based on discovering distinctly different kinds of solutions than GRPO. Appendix A.3 presents additional experiments with varying $\alpha$ and $\beta$ values, yielding similar conclusions.

**ES is more robust against reward hacking.** GRPO with $\beta = \{0.0, 0.01\}$ sometimes hacked the reward, that is, produced responses that were short but contain nonsensical symbols rather than words. By increasing the KL-penalty via higher $\beta$ values, reward hacking could be prevented. The optimal $\beta$ is likely to be problem specific and to require extensive search to find. In contrast, ES does not receive any feedback about the divergence of the fine-tuned model, and only seeks to optimize conciseness. Regardless, it did not exhibit any reward hacking, despite achieving mean reward comparable to GRPO with $\beta = \{0.0, 0.01\}$. This result again suggests that ES finds a different way of optimizing the reward function.

**ES fine-tuning is reliable across runs.** Fine-tuning LLMs is computationally expensive, and it is therefore critical that it leads to consistent results across runs. Table 2 presents the mean and standard deviation of the conciseness reward and KL divergence across four independent runs after $1,000$ iterations. A mean reward cut-off of $> 0.85$ was used to down-select hyperparameter combinations, ensuring that only the best ES and GRPO configurations were included in the analysis.

As shown in Table 2, ES achieved consistent conciseness rewards, indicated by a low reward standard deviation (0.004 and 0.008) over four runs with different random seeds. GRPO has $15.5\times$ higher standard deviation (0.041–0.062), suggesting that its results were much less consistent. The results on KL divergence were similar. For instance, while ES ($\sigma = 0.0015$) and GRPO ($\beta = 0.0167$) achieved similar mean rewards, GRPO exhibits a $1.95\times$ higher KL divergence mean and $3.83\times$ greater standard deviation. Similarly, while ES ($\sigma = 0.001$) achieves a slightly lower reward compared to GRPO ($\beta = 0.0167$), GRPO ($\beta = 0.0167$) has a $5.8\times$ higher KL divergence and a $8.44\times$ higher standard deviation. Thus, ES fine-tuning is more reliable than GRPO.

## 5 DISCUSSION AND FUTURE WORK

Exploration in parameter space plays a key role in the surprisingly good fine-tuning performance of ES. As discussed by Rückstieß et al. (2010) and Plappert et al. (2018), sampling noise in parameter space ensures that the entire action trajectory, i.e., the sequence of tokens, only depends on one single sampling, leading to significantly lower variance in rollouts, i.e., in response generation. As a result, gradient estimation is more reliable and convergence is more stable. In contrast, action space exploration in RL injects noise at every step, i.e., at each token position, resulting in high variance in the sequence generation. The behavior of RL therefore is much less reliable than ES, as was seen in Table 2. Moreover, step-wise exploration in action space promotes reward hacking by increasing the chance of sampling a single hacking action. One example is the nonsensical symbol sampled during RL that can hack the conciseness reward.

Another key difference between ES and RL is that ES intrinsically optimizes a solution distribution (Lehman et al., 2018), while RL optimizes a single solution. This property makes it more difficult for ES to hack the reward since a single hacked solution usually does not have a high-quality solution distribution around it. This property also results in solutions that are more robust to noisy

perturbations in parameter space (Lehman et al., 2018), making them more robust to adversarial attacks and less likely to be compromised in other follow-up fine-tuning tasks (Chen et al., 2025).

In the experiments in this paper, extensive hyperparameter tuning was performed for RL methods, resulting in specific RL hyperparameters for different model sizes and families. In comparison, ES was found to be less sensitive to hyperparameters, and the same set of hyperparameters was used for all experiments. While there are many common enhancements for ES (Salimans et al., 2017), none were used in the experiments so that the power of vanilla ES could be clearly demonstrated. Thus, it may be possible to improve the results with more extensive hyperparameter tuning and other enhancements.

One counterintuitive result is that the ES implementation only needs a population of 30 to effectively optimize billions of parameters. In contrast, previous work (Salimans et al., 2017; Zhang et al., 2017; Lehman et al., 2018; Lorenc & Neruda, 2025) used populations of 10,000 or more for models with millions or fewer parameters. An interesting future direction is to analyze how such small populations are possible. Perhaps this is related to the observed low intrinsic dimensionality of LLMs (Aghajanyan et al., 2021). Another promising direction is to use ES to perform unsupervised fine-tuning based on internal behaviors of LLMs, such as confidence calculated based on semantic entropy and semantic density (Qiu & Miikkulainen, 2024; Farquhar et al., 2024). Such fine-tuning cannot be done with RL, since action space exploration does not change the internal representations of LLMs (that is, each action sampling is generated via output distribution without changing the internal parameters). In a broader sense, since ES does not need process rewards during exploration, it may be a necessary ingredient for superintelligence (Mucci & Stryker, 2023), which would be difficult to achieve by supervised learning using process guidance from human data. Massive parallelization of ES will speed up exploration by distributing the computations across GPU machines or even data centers.

An important question is: what are the underlying computational mechanisms that make ES and RL behave so differently? While this question requires significant further work, a possible hypothesis emerges from the experiments in this paper. Many fine-tuning objectives, like conciseness and the Countdown task, are long-horizon outcome-only objectives. The reward signal is jagged, making it difficult to navigate with gradient-based post-training methods. RL and ES both provide workarounds via effective noise injection to "smooth out" the jagged reward landscape. In the case of RL, noise is introduced from Monte-Carlo sampling of each token during a rollout, averaged over many rollouts, which effectively smooths the sampling process but does not necessarily guarantee that the reward landscape is smooth in parameter space. RL's gradient estimation therefore has a high-variance, and its signal-to-noise ratio becomes worse with longer sequences and sharper policies (i.e. those with lower entropy), and therefore prone to undesirable outcomes such as reward hacking.

In contrast, ES injects noise directly into the parameter space via explicit Gaussian convolution, which effectively smooths out the jagged reward landscape. As a result, it provides a more stable way of exploring the landscape, leading to more consistent, efficient, and robust optimization (as observed in the experiments and in Appendix A.5). Moreover, the larger the models and the sharper the policies, the more jagged the reward landscapes; therefore, ES is likely to have an advantage in fine-tuning them. Direct evidence for this hypothesis still needs to be obtained, but it provides a plausible mechanistic explanation, and a direction for future work. Eventually such work could result in better fine-tuning methods, as well as an improved understanding of LLMs in general.

## 6 CONCLUSION

This paper introduces and evaluates a new approach to fine-tuning LLMs, based on scaling up ES to billions of dimensions. The approached performed significantly better than the standard RL fine-tuning in the Countdown task, which has sparse long-horizon rewards. ES was found to be more sample efficient, less sensitive to hyperparameter setup, and to achieve consistently better results across multiple LLMs. Further empirical studies on fine-tuning for conciseness revealed that ES is less likely to hack the reward, and behaves reliably across multiple runs. The mechanisms underlying these differences still need to be characterized, but a plausible hypothesis is that the exploration in ES is better suited for the jagged reward landscapes in large models. ES therefore constitutes a promising alternative to RL in fine-tuning LLMs.

## REPRODUCIBILITY STATEMENT

The experimental and parametric setup is provided in full details in Appendix A.1 for reproducing all the experimental results reported in this paper. Source codes for generating the experimental results are included in the supplementary material.

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

# A APPENDIX

## A.1 EXPERIMENTAL SETUP

**Experimental setup for the Countdown experiments.** Representative models from the Qwen2.5 family (0.5B–7B) and the LLaMA3 family (1B–8B) were fine-tuned for this task. For the PPO experiments, a grid search was first performed around common hyperparameter settings and the best-performing values used (Table 3). For the GRPO experiments, a grid search was performed around the settings of Pan et al. (2025) and the best-performing values used. GRPO experiments were run with two different group sizes: $N = 8$, following the common practice in GRPO training for the Countdown task, and $N = 30$, aligning with the population size in ES. For all the ES, GRPO and PPO, the total number of sample evaluations was the same. The ES population size was $N = 30$, noise scale $\sigma = 0.001$, and learning rate $\alpha = 5 \times 10^{-4}$ across all experiments. To evaluate accuracy, a set of 200 samples were used during training, and a different set of 2000 samples during testing. For ES, results were reported on the test set after training for 500 iterations. For RL, the training was stopped after the same total number of sample evaluations as in the ES runs. An example of the prompt and the response is provided in Appendix A.2.

| Method | Model | (1e−3, 1e−6) | (1e−3, 1e−5) | (5e−3, 1e−6) | (5e−3, 1e−5) |
|---|---|---|---|---|---|
| | Qwen-0.5B-Instruct | ✓ | | | |
| | Qwen-1.5B-Instruct | ✓ | | | |
| | Qwen-3B-Instruct | ✓ | | | |
| PPO | Qwen-7B-Instruct | | ✓ | | |
| | LLaMA-1B-Instruct | | ✓ | | |
| | LLaMA-3B-Instruct | | | | ✓ |
| | LLaMA-8B-Instruct | | | ✓ | |
| | Qwen-0.5B-Instruct | | ✓ | | |
| | Qwen-1.5B-Instruct | | | ✓ | |
| | Qwen-3B-Instruct | | ✓ | | |
| GRPO | Qwen-7B-Instruct | ✓ | | | |
| | LLaMA-1B-Instruct | | | | ✓ |
| | LLaMA-3B-Instruct | | ✓ | | |
| | LLaMA-8B-Instruct | ✓ | | | |

Table 3: Hyperparameter Sweep across Models under PPO and GRPO. Each pair $(\cdot, \cdot)$ denotes (KL-divergence penalty coefficient $\beta$, learning rate $\alpha$); the label '✓' indicates the best hyperparameter setting for each model-method combination.

**Experimental setup for the Conciseness experiments.** In each experiment, Qwen-2.5-7B-Instruct (Yang et al., 2025) was fine-tuned using both ES and GRPO and evaluated using a held-out evaluation set. Each run was repeated four times, using a different random seed each time. For each GRPO experiment, the group size $N = 30$, and learning rate $\alpha = 5 \times 10^{-6}$. Ten log-spaced values from 0.01 to 1.0 were evaluated for the the KL-divergence penalty coefficient $\beta$, as well as $\beta = 0.0$. Appendix A.3 presents additional experiments with varying $\alpha$ and $\beta$ values. For ES, the population size $N = 30$, ensuring that GRPO and ES generated the same number of responses per prompt, resulting in the same training exposure. Models were fine-tuned with $\sigma = \{0.0005, 0.001, 0.0015\}$, with a learning rate $\alpha = \frac{\sigma}{2}$. Both GRPO and ES experiments were run for $1,000$ iterations, and a checkpoint saved every 200 iterations. Table 4 shows the dataset of prompts and verifiable solutions used during fine-tuning; note that it consists of only two examples. Similarly, Table 5 lists the prompts and verifiable solutions used in evaluating each fine-tuned model.

## A.2 EXAMPLES OF COUNTDOWN TASK OUTPUTS

Figure A.2 shows an example of the Qwen2.5-3B model's output on the Countdown task after 400 ES iterations. In order to improve accuracy, the model learned to adjust its formulas by comparing the generated answer with the correct one, and to verify its own answers for correctness.

| Prompt | Verifiable Solution |
|---|---|
| Solve: $3 + 5 =$ | 8 |
| If all birds can fly and penguins are birds, can penguins fly | No |

Table 4: Prompts and verifiable solutions used in fine-tuning the models for conciseness. Two examples is enough to achieve this goal.

| Prompt | Verifiable Solution |
|---|---|
| What is the capital of France? | Paris |
| Calculate: $12 \times 7 =$ | 84 |
| Is the statement "All cats are mammals" true or false? | True |
| What comes next in the sequence: $2, 4, 6, 8, ?$ | 10 |
| Translate "Hello" to Spanish: | Hola |
| What is $15\%$ of 200? | 30 |
| Name one primary color: | Red |
| How many days are in a week? | 7 |

Table 5: Prompts and verifiable solutions used to evaluate the fine-tuned models. More examples are necessary than during fine-tuning to make the evaluation reliable.

---

**Iteration 400**

**[1] Input:**

You are a helpful assistant. You first think about the reasoning process in your mind and then provide the user with the answer. Using the numbers [28 3 4 52], create an equation that equals 44. You can use basic arithmetic operations (+, −, *, /) and each number can only be used once. Show your work in `<think> ... </think>` tags. And return the final answer in `<answer> ... </answer>` tags, for example `<answer> (1 + 2) / 3 </answer>`.

**[1] Response:**

`<think>` First, let's consider all possible combinations of the numbers 49, 41, and 73 using basic arithmetic operations. We need to use each number exactly once and combine them to get 17.
1. Start with simple addition and subtraction: - 49 + 41 - 73 = 80 - 73 = 7 (too high) - 49 - 41 + 73 = 8 + 73 = 81 (too high) - 73 - 49 - 41 = 24 - 41 = -17 (too low)
2. Try multiplication and division: - (49 * 41) / 73 = 2029 / 73 ≈ 27.83 (not close to 17) - (73 * 41) / 49 = 3013 / 49 ≈ 61.47 (too high) - (49 * 73) / 41 = 3637 / 41 ≈ 88.80 (too high) - (41 * 73) / 49 = 3013 / 49 ≈ 61.47 (same as above)
3. Check different orders: - 73 - (49 + 41) = 73 - 90 = -17 (too low) - 49 - (73 - 41) = 49 - 32 = 17 (this looks promising)
So, the equation 49 - (73 - 41) equals 17. Let's verify it: - Calculate inside the parentheses first: 73 - 41 = 32 - Then subtract from 49: 49 - 32 = 17
This satisfies our requirement. `</think> <answer>` 49 - (73 - 41) `</answer>`
Reward: 1.1000 (Format: 1.0000, Answer: 1.0000)

Figure 2: An example of a countdown task interaction.

---

A.3 EXTENDED CONCISENESS EXPERIMENTS

In this section, the conciseness experiments are extended to investigate the impact of different learning rates on GRPO training.

**GRPO with different learning rates.** Further GRPO experiments were run over four seeds with $\beta = \{0, 0.01, 0.1, 1.0\}$, varying the learning rate $\alpha = \{2 \times 10^{-6}, 3 \times 10^{-6}, 4 \times 10^{-6}, 5 \times 10^{-6}\}$. A total of 20 responses were sampled per evaluation prompt. Figure 3a shows the mean reward and KL

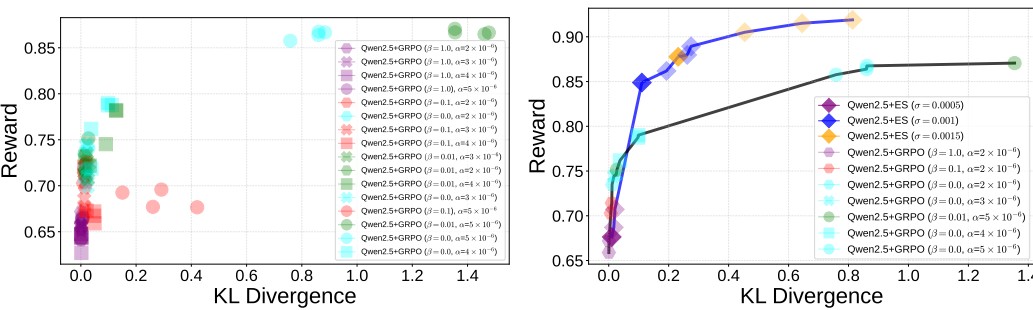

(a) GRPO models results over various learning rates.        (b) ES and GRPO Pareto fronts.

Figure 3: GRPO behavior with different learning rates. (a) GRPO models trained using different learning rates and $\beta$ values. Both conciseness reward and KL divergence increase with higher learning rates. (b) The ES Pareto front (blue line, top) plotted with the GRPO Pareto front (black line, bottom) over different model learning parameters. ES dominates GRPO across the whole range.

divergence of each fine-tuned model. As the learning rate increases, both mean reward and mean KL divergence increase. The best models with respect to reward are trained using $5 \times 10^{-6}$ and $\beta = \{0.0, 0.01\}$, obtaining rewards greater than $0.85$. Figure 3b further displays the GRPO Pareto front (black line, bottom) across these learning rates, comparing it with the ES Pareto front (blue line, top). The majority of Pareto optimal models across these learning rates obtain a mean reward of less than $0.8$ and a KL divergence of less than $0.4$. The ES Pareto front dominates that of GRPO over different learning rates and $\beta$ values.

Next, the reward distribution for each $\alpha$ and $\beta$ value for GRPO was compared with that of ES, starting with learning rates $2 \times 10^{-6}$ and $3 \times 10^{-6}$. Figures 4a and Figure 4b show that all GRPO models stay close to the Qwen2.5-7B-Instruct base model reward distribution, despite the variation in $\beta$. In contrast, ES shifts the reward distribution to the right with a density peak around $1.0$, i.e. towards higher rewards. The learning rate was then further increased to $4 \times 10^{-6}$ (Figure 4c). As a result, for $\beta = 0.0$ and $\beta = 0.01$, GRPO shifts the reward distribution to the right towards higher rewards. However, they are still lower than those of ES. As the learning rate is increased further to $5 \times 10^{-6}$ (Figure 4d), GRPO is sufficiently able to optimize the reward: with $\beta = 0.0$ and $\beta = 0.01$, it peaks around $1.0$. Thus, high learning rate combined with low $\beta$ is important for GRPO to optimize the reward. However, as was discussed before, such a setting often breaks the performance of the model.

### A.4 Training Curves and Accuracy Improvement of ES and RL on the Countdown Task

As shown in Figure 6, ES consistently outperformed RL across all tested models throughout training. On smaller models such as Qwen2.5-0.5B and Llama-3.2-1B, RL showed almost no improvement, whereas ES steadily increased accuracy. On mid-sized models like Qwen2.5-1.5B, Qwen2.5-3B and Llama-3.2-3B, ES achieved substantial gains, reaching accuracy levels that RL never approached even in extended training. On larger models such as Qwen2.5-7B and Llama-3.1-8B, RL improved more than in the smaller models, but ES still maintained a clear and consistent advantage, achieving the highest accuracy throughout. In addition, as shown in Figure 5, we compute the relative improvements of PPO, GRPO, and ES over their respective base models across different model families. ES delivers the consistently largest improvements in all cases.

### A.5 Parameter Magnitude Shifts by Evolutionary fine-tuning

This section characterizes how parameter magnitudes changed in ES fine-tuning in the countdown and conciseness experiments. Specifically, Figures 7 and 8, left column, show histograms of the absolute parameter magnitude shifts $\Delta$ before and after finetuning Llama and Qwen models, overlaid with random walk, on the Countdown task reported in Table 1. The right column in these figures shows the difference between $\Delta$ and the random walk.

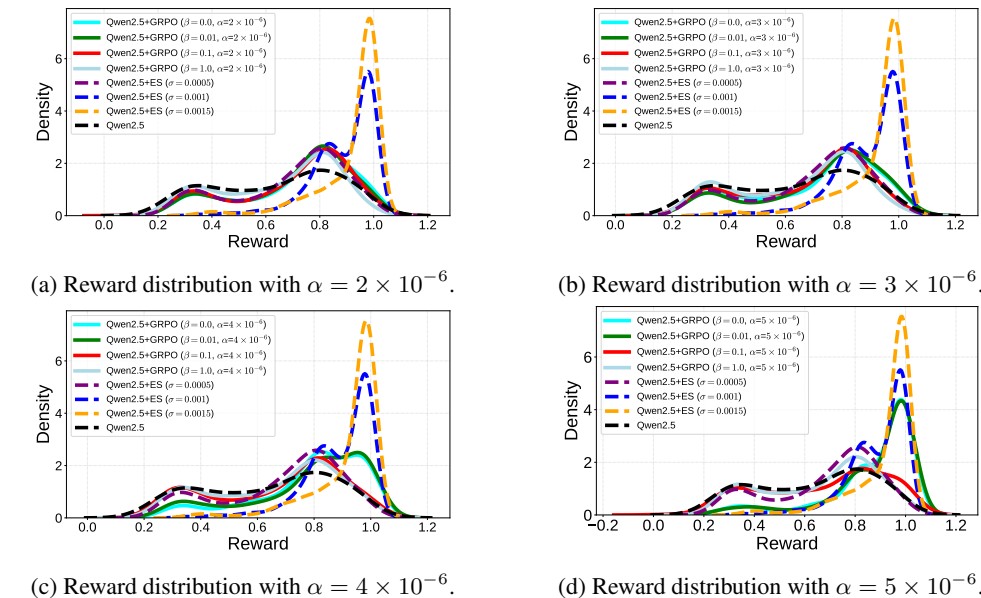

(a) Reward distribution with $\alpha = 2 \times 10^{-6}$.

(b) Reward distribution with $\alpha = 3 \times 10^{-6}$.

(c) Reward distribution with $\alpha = 4 \times 10^{-6}$.

(d) Reward distribution with $\alpha = 5 \times 10^{-6}$.

Figure 4: Reward distributions in fine-tuning for conciseness with different learning rates $\alpha = \{2 \times 10^{-6}, 3 \times 10^{-6}, 4 \times 10^{-6}, 5 \times 10^{-6}\}$ and $\beta = \{0.0, 0.01, 0.1, 1.0\}$ compared to ES on the Qwen2.5-7B-Instruct base model. Whereas GRPO distribution is similar to the base model, ES shifts it to the right, i.e. higher rewards. Higher rewards can only be achieved with GRPO with high learning rates and low $\beta$, which setting often breaks to model's performance.

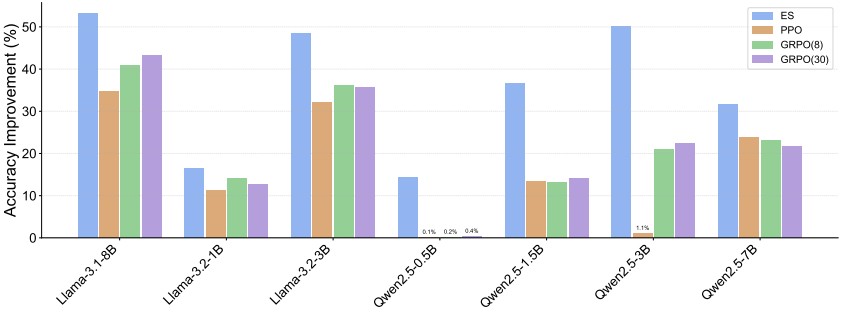

Figure 5: Accuracy Improvement over Base Models with ES vs RL across Model Families. ES results in consistently largest improvements in all cases.

For most models, $\Delta$ deviates very little from random walk. This is a counterintuitive result since fine-tuning actually resulted in a significant performance boost. A closer inspection reveals that most of the deviation was concentrated around zero. A likely explanation is that there are precision issues around zero, particularly with small bin sizes, which may lead to such deviations.

More significantly, a systematic deviation from the random walk was observed in conciseness fine-tuning of the largest model, Qwen2.5-7B-Instruct (Figure 9). The distribution shifts toward abundant small magnitude edits, suggesting that small parameter tweaks may be most significant in influencing output behavior. This result reinforces observations in prior studies (e.g. Liu et al., 2025). A possible explanation is that large models encode functionality in a more redundandant manner, and therefore minor tweaks are sufficient to achieve fine-tuning objectives. In fact, the changes are nearly indistinguishable from random walk in Figures 7 and 8 likely because they are benevolent wrt. the fine-tuning objective. A more thorough investigation of these hypotheses is a most interesting direction of future work, potentially resulting in a better understanding of fine-tuning and information processing principles in LLMs in general.

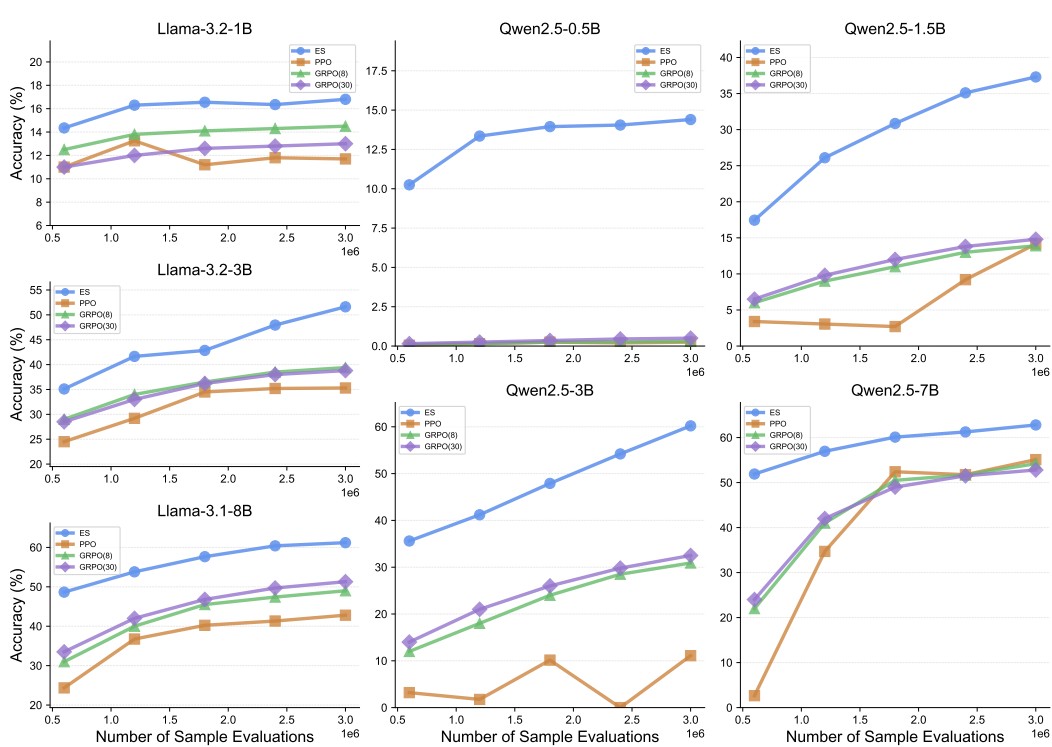

Figure 6: Training curves of ES and RL across two model families and six sizes in the countdown task. ES fine-tuning results in significantly better performance in all cases. It is able to improve even the smallest model where RL methods are ineffective. ES is also more sample efficient than RL: in most cases, it only needs less than 20% of the training sample evaluations of RL to achieve similar performance.

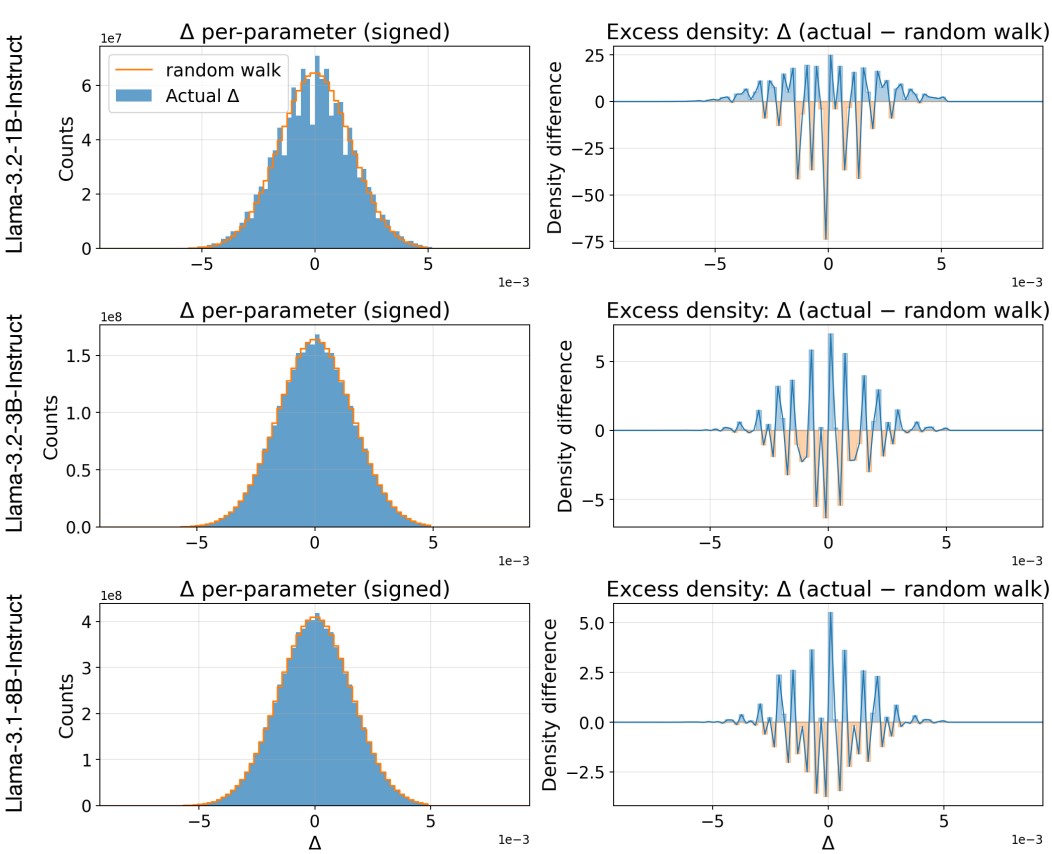

Figure 7: Parameter magnitude shift histograms for the Countdown task in Llama models optimized by ES. The changes are similar to those of a random walk, concentrated around zero, likely due to numerical inaccuracies.

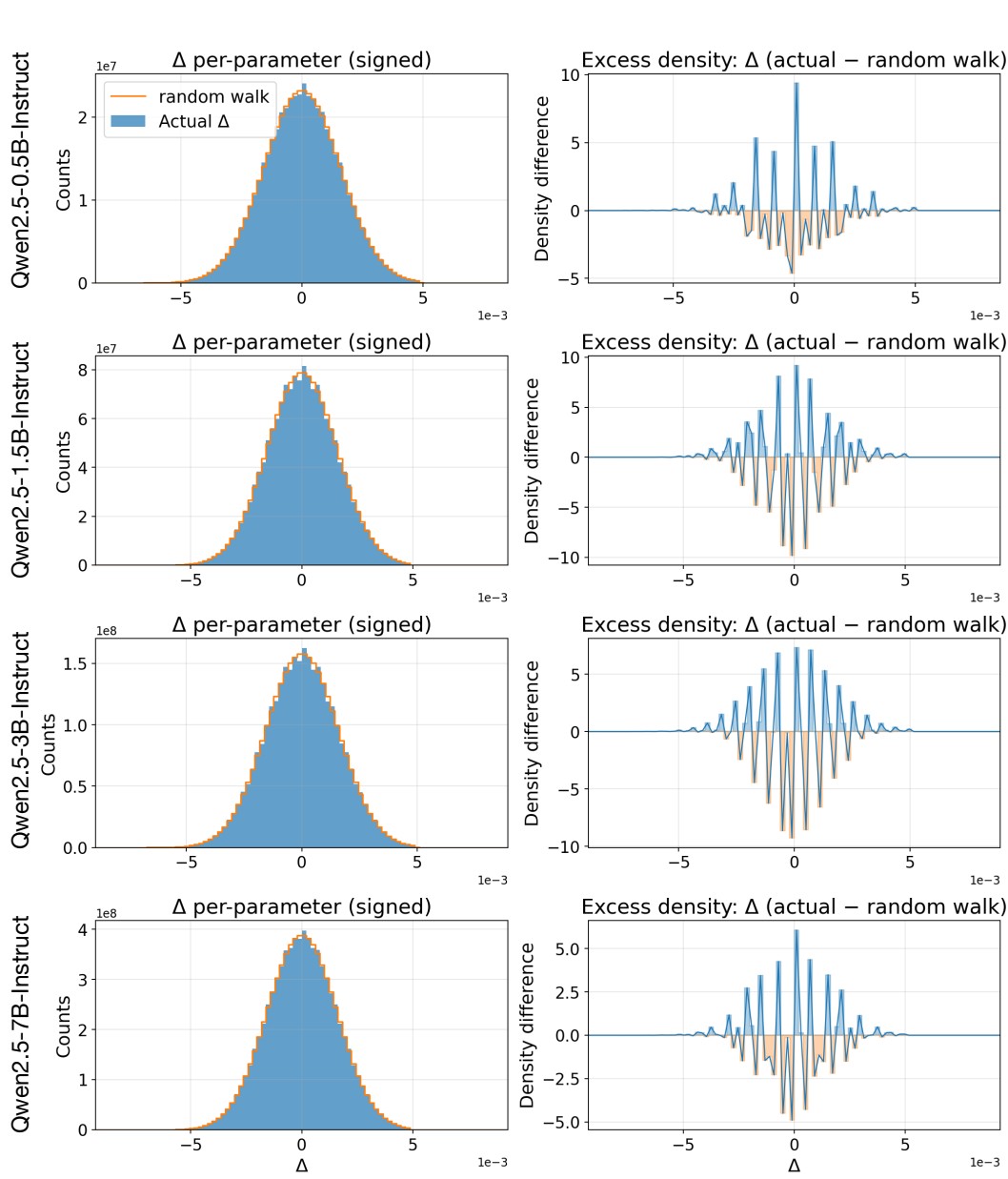

Figure 8: Parameter magnitude shift histograms for the Countdown task in Qwen models optimized by ES. The results are consistent with those observed in Llama models.

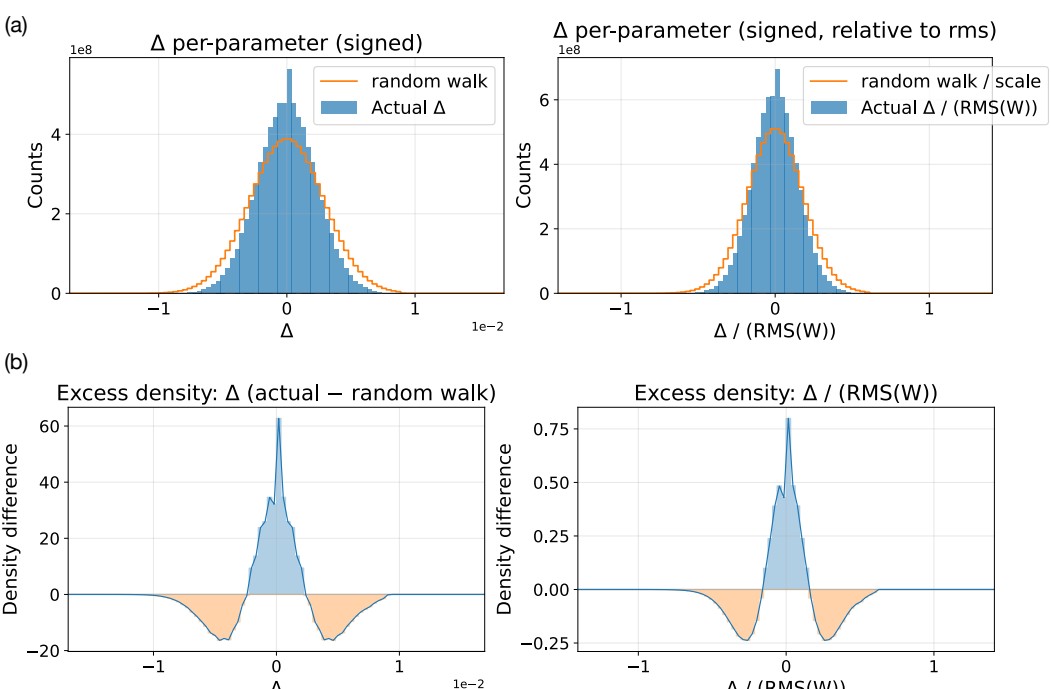

Figure 9: Parameter magnitude shift histograms in conciseness fine-tuning in Qwen2.5-7B-Instruct model with ES. In this case, the model is large and the fine-tuning goals is different, revealing a potentially significant pattern of primarily small changes. The hypothesis (to be analyzed more thoroughly in future work) is that behavior is coded in large models in a redundant manner, making it possible to achieve this fine-tuning objective through numerous small changes.

