# OpenReview forum: "Evolution Strategies at Scale: LLM Fine-Tuning Beyond Reinforcement Learning"
_ICLR.cc/2026/Conference — ICLR 2026 Conference Withdrawn Submission_

### Official Review · Reviewer_psfV · 2025-10-26

**Soundness:** 3
**Presentation:** 3
**Contribution:** 3
**Rating:** 4
**Confidence:** 4

**Summary:**

This paper presents a scalable application of Evolutionary Strategies (ES) for RL–based fine-tuning of LLMs. The method is both memory-efficient and easily parallelizable, maintaining the core advantages of ES. Experiments on the Countdown dataset demonstrate that ES outperforms existing RL methods across a range of base models, while also achieving greater sample efficiency. Further evaluation on the Conciseness task shows that ES is Pareto-superior to RL approaches—offering improved stability and a reduced tendency toward reward hacking.

**Strengths:**

1. The application of ES to LLMs is very interesting and has a large potential impact on the field.

2. The paper presents many noteworthy results, such as: 1) exploration in the parameter space is effective when the base models are small/weak and 2) ES results in models that are close to the base model w/o any explicit regularization.

3. The paper is well written and very easy to follow.

**Weaknesses:**

1. My primary concern is the limited generality of the results. Although the experiments are interesting, they are confined to the Countdown and (a toy) Conciseness task. It remains unclear how ES would perform on established datasets such as MATH, GSM8K etc and whether their findings (improved perf, sample efficiency) still hold.

2. In Section 3.2, it is unclear which components of the ES method are standard or derived from prior work (presumably 1, 2, and 4), and which ones represent the novel contributions of this paper (likely 3, 5, 6, and 7). Clarifying this would improve the presentation and make the paper’s contributions more transparent.

**Questions:**

1. How does ES perform on datasets such as MATH, GSM8k etc?

Overall, the results are promising; however, I am concerned that the findings may be specific to the Countdown task and may not generalise to other reasoning benchmarks. I would be happy to raise my score if additional results were provided demonstrating ES performance on standard reasoning datasets.

---

### Official Review · Reviewer_dhcs · 2025-10-27

**Soundness:** 1
**Presentation:** 2
**Contribution:** 1
**Rating:** 2
**Confidence:** 4

**Summary:**

In this paper, the author claims to scale up a perturbation based ES (Evolution Strategy) to the scale of multi-billion and use it for fine tuning LLMs. The scale up focuses on making execution in parallel (on GPU) feasible and faster . The experiments are done in the count down tasks, and compared to PPO and GRPO.

**Strengths:**

Scaling up ES for LLM is an important task. FUrther more, Details in the analysis of the one tasks is well presented (although unfortunately one task is not enough, see weaknesses below.)

**Weaknesses:**

There are several serious weaknesses that, in my view, warrant rejection of this manuscript.

The main claim is that ES is better than RL for LLM fine-tuning. However, the experiments presented do not sufficiently support this claim.

Only one task, the countdown task, is used to justify the conclusion. Unfortunately, showing superior performance on a single, relatively simple task is not persuasive. To make a convincing argument, the authors should evaluate across a broader range of tasks. For example, representative benchmarks could include language understanding tasks such as MMLU-Redux (https://arxiv.org/abs/2406.04127), MMLU-Pro (https://arxiv.org/abs/2406.01574), SimpleQA (https://openai.com/index/introducing-simpleqa/), and HellaSwag-Pro (https://arxiv.org/abs/2502.11393v1); coding tasks such as SWE-bench (https://openai.com/index/introducing-swe-bench-verified/) and LiveCodeBench (https://arxiv.org/abs/2403.07974); and math tasks such as MATH-500 (https://github.com/openai/prm800k/tree/main?tab=readme-ov-file#math-splits). At a minimum, representative tasks from these categories should be included for comparison. In its current form, relying on a single task is far from sufficient to support the paper’s central claim.

From an RL-for-LLM standpoint, the manuscript’s comparison of ES is limited to only a few RL methods (PPO and GRPO). This omits a broad range of relevant approaches, both classical and recent. Notably, recent methods such as KTO (https://arxiv.org/abs/2402.01306), NCA (https://arxiv.org/abs/2402.05369), DPO (https://arxiv.org/abs/2305.18290) and its improvements like PRO (https://arxiv.org/abs/2505.23316), SimPO (https://arxiv.org/abs/2405.14734), and ORPO (https://arxiv.org/abs/2403.07691) are not considered. Including comparisons with at least some of these recent methods is necessary to make the evaluation credible and complete.

From an ES perspective, the method presented is based on a simple form of evolutionary strategy using naïve Gaussian perturbations. While this approach is acceptable as a baseline, neglecting more advanced ES techniques raises several unanswered questions. For instance, methods such as PGPE (https://people.idsia.ch/~juergen/icann2008sehnke.pdf), Natural PGPE (https://proceedings.neurips.cc/paper_files/paper/2010/file/44c4c17332cace2124a1a836d9fc4b6f-Paper.pdf), and Policy Gradients with Structured Control Variates (https://arxiv.org/abs/1906.08868) could be discussed, serving as either stronger foundations for implementing efficient ES approaches suitable for LLM fine-tuning, or a case showing the issues in these advances when used for large scale models LLMs

**Questions:**

N/A

---

### Official Review · Reviewer_5mn2 · 2025-11-01

**Soundness:** 2
**Presentation:** 3
**Contribution:** 3
**Rating:** 4
**Confidence:** 3

**Summary:**

This paper proposes scaling evolution strategies to finetune small LMs (1B-8B params). The authors implement an ES algorithm that perturbs parameters of the lm directly, rather than actions. They then compare it against PPO and GRPO on 2 tasks: the Countdown symbolic reasoning benchmark and a conciseness fine-tuning task.

**Strengths:**

1. Overall, I really liked the engineering contributions and some clever tricks (like layer-wise perturbation, seed-based noise retrieval, etc.) applied to make ES stable for billion-parameter models.
2. I also appreciate the authors running the tasks on different model families (Qwen, LLaMA) and multiple sizes (0.5B-8B).
3. The reward-KL tradeoff analysis in Figure 1 and Section 4.2 provide good intuition about behavioral differences.
4. The finding that ES works on smaller models where RL fails (like qwen-0.5B: 0.1%→14.4%) is interesting.

**Weaknesses:**

1. I feel that showing result on one neurosymbolic task is insufficient to make a broad claim that ES benefits. The main claim is that ES provides "a new direction in LLM fine-tuning beyond what current RL techniques provide." But we know Countdown is not representative of real LLM fine-tuning benchmarks and there's no evaluation on standard benchmarks like AlpacaEval or in general, baselines on any summarization, code generation, or mathematical reasoning domain.
2. The discussion of why a small population size suffices is speculative and not supported by any experiments (e.g., no study of population size vs. performance).  Also, [minor] the authors claim ES avoids reward hacking but don't report what percentage of RL runs did such hacking.
3. I feel that the abstract and introduction sometimes overstate the generality and impact of the results. The paper does not sufficiently discuss limitations or scenarios where ES may not be suitable.
4. On page 3, the authors say "outperforms all baselines" but I don't see any baselines like MeZO, or CMA-ES included in the comparisons.

Overall, while I think this paper has some new ideas and promising results, a lot of the text in the current submission feels highly overstated to me.

**Questions:**

1. How does the wall-clock time and total compute cost of ES compare to RL methods, especially as model size increases?
2. How sensitive is ES performance to the choice of population size? Is there a lower bound for effective optimization at scale?
3. Can the authors provide more evidence (or counterexamples) regarding ES’s resistance to reward hacking, especially on more complex tasks?
4. Are there tasks or reward structures where ES is not suitable?

---

### Note · Authors · 2025-12-03

**Comment:**

We would like to thank the reviewers for their time and effort in reviewing our paper. We are currently running experiments on more benchmarks, including more math reasoning problems and other types of tasks. We are also adding more baselines, including new variants of GRPO and new RL fine-tuning algorithms. Considering the time limitation during the rebuttal phase, we decided to withdraw so that we can have more time to complete all the experiments in a conclusive manner. We will add all the experimental results to our next version.

**Withdrawal Confirmation:**

I have read and agree with the venue's withdrawal policy on behalf of myself and my co-authors.